

# Influence of rimonabant treatment on peripheral blood mononuclear cells; flow cytometry analysis and gene expression profiling

Stefan Almestrand[1,*], Xiao Wang[1,**], Åsa Jeppsson-Ahlberg[2], Marcus Nordgren[1,***], Jenny Flygare[1,****], Birger Christensson[1], Stephan Rössner[3] and Birgitta Sander[1]

[1] Department of Laboratory Medicine, Division of Pathology, Karolinska Institutet and Karolinska University Hospital Huddinge, Stockholm, Sweden
[2] Pathology/Cytology, Karolinska University Hospital Huddinge, Stockholm, Sweden
[3] Department of Medicine, Karolinska University Hospital Huddinge, Stockholm, Sweden
[*] Current affiliation: AstraZeneca, Södertälje, Sweden
[**] Current affiliation: Center for Primary Health Care Research, Skåne University Hospital, Malmö, Sweden
[***] Current affiliation: Laboratory of Lipid Biochemistry and Protein Interactions, Department of Cellular and Molecular Medicine, KU Leuven, Leuven, Belgium
[****] Current affiliation: Department of Laboratory Medicine, Division of Clinical Chemistry, Karolinska Institutet, Stockholm Sweden

Corresponding author
Birgitta Sander,
birgitta.sander@ki.se

## ABSTRACT

The cannabinoid receptor type 1 (CB1) antagonist rimonabant has been used as treatment for obesity. In addition, anti-proliferative effects on mitogen-activated leukocytes have been demonstrated *in vitro*. We have previously shown that rimonabant (SR141716A) induces cell death in *ex vivo* isolated malignant lymphomas with high expression of CB1 receptors. Since CB1 targeting may be part of a future lymphoma therapy, it was of interest to investigate possible effects on peripheral blood mononuclear cells (PBMC) in patients treated with rimonabant. We therefore evaluated leukocyte subsets by 6 color flow cytometry in eight patients before and at treatment with rimonabant for 4 weeks. Whole-transcript gene expression profiling in PBMC before and at 4 weeks of rimonabant treatment was done using Affymetrix Human Gene 1.0 ST Arrays. Our data show no significant changes of monocytes, B cells, total T cells or T cell subsets in PBMC during treatment with rimonabant. There was a small but significant increase in CD3−, CD16+ and/or CD56+ cells after rimonabant therapy. Gene expression analysis detected significant changes in expression of genes associated with innate immunity, cell death and metabolism. The present study shows that normal monocytes and leukocyte subsets in blood remain rather constant during rimonabant treatment. This is in contrast to the induction of cell death previously observed in CB1 expressing lymphoma cells in response to treatment with rimonabant *in vitro*. These differential effects observed on normal and malignant lymphoid cells warrant investigation of CB1 targeting as a potential lymphoma treatment.

## INTRODUCTION

The endocannabinoid system consists of the cannabinoid type 1 (CB1) and cannabinoid type 2 (CB2) receptors, their endogenous ligands anandamide and 2-arachidonoyl glycerol and the enzymes involved in their biosynthesis and metabolism (*Di Marzo, Bifulco & De Petrocellis, 2004*). CB1 is involved in the regulation of food intake, energy balance and metabolism of glucose and lipids (*Di Marzo & Matias, 2005*). In clinical studies, CB1 receptor blockage by the selective CB1 antagonist rimonabant (SR141716A) induced weight loss and improvement in serum lipid, glucose and insulin levels by targeting central and peripheral CB1 receptors (*Van Gaal et al., 2008*). However, some patients experienced depression, and this was considered an unacceptable side effect for treating obesity/metabolic syndrome. Hence, the drug was withdrawn from clinical use but there is remaining interest in some of its many potential medical applications (*Cooper & Regnell, 2014*; *Zhou et al., 2012*) including treatment of various malignancies. It is therefore of interest to investigate possible adverse effects on blood cells in patients treated with rimonabant.

The endocannabinoid system is regulating various aspects of lymphocyte proliferation, maturation and immune response (*Klein, 2005*; *Muppidi et al., 2011*; *Pandey et al., 2009*; *Pereira et al., 2009*; *Sido, Nagarkatti & Nagarkatti, 2014*). Targeting the endocannabinoid system may therefore be a possible new treatment option in various lymphoproliferative disorders. CB1 receptors are expressed on cells of the immune system, but generally at lower levels than CB2 (*Bouaboula et al., 1993*; *Galiegue et al., 1995*). We and others have found that CB1 and CB2 are highly expressed on neoplastic lymphocytes in malignant lymphoma (*Gustafsson et al., 2008*; *Islam et al., 2003*; *McKallip et al., 2002*; *Wasik et al., 2014*). Targeting of CB1 and CB2 with endogenous or synthetic agonists reduced cell proliferation *in vitro* and *in vivo* and induced programmed cell death selectively in tumor cells of mantle cell lymphoma (*Flygare et al., 2005*; *Gustafsson et al., 2006*; *Gustafsson et al., 2008*; *Wasik et al., 2011*). Similarly, CB2 agonists induced cell death in T cell lymphoblastic leukemia (*McKallip et al., 2002*). Also the CB1 antagonist rimonabant impaired proliferation and induced cell death in *ex vivo* isolated mantle cell lymphoma cells, alone, or in combination with anandamide (*Flygare et al., 2005*). Others have reported antiproliferative effects of rimonabant on *in vitro* activated PBMC but not on freshly isolated, non-activated PBMC (*Gallotta et al., 2010*; *Malfitano et al., 2008*). These results show that CB1 blockade may have immunomodulatory and antiproliferative effects on malignant lymphoma and on activated normal lymphocytes *in vitro* but seems to spare resting lymphocytes. Very little is published on the effects of rimonabant on human PBMC *in vivo*, and the aim of this study was to investigate how treatment with rimonabant affected blood leukocytes. We therefore collected blood cells from obese patients treated with rimonabant and analyzed blood leukocytes by flow cytometry before and during treatment. To investigate which genes were differentially expressed in PBMC during rimonabant treatment, we used oligonucleotide arrays to compare gene expression profiles in PBMC before and at 4 weeks of treatment. This pilot study shows that rimonabant treatment induces expression of genes involved in immune responses but have only marginal effects on leukocyte subset frequencies in blood.

## MATERIALS AND METHODS

### Patients and study design

Rimonabant was prescribed to eight patients, admitted to the Overweight Study Unit at the Department of Medicine, Karolinska University Hospital. Rimonabant was administered according to the manufacturers guidelines. All patients had a BMI >35 kg/m$^2$, were treated on clinical indications (metabolic and mechanical disability) and without mental disturbances. They were not included in any other study. The clinical characteristics of these patients are presented in Table 1. Blood samples were collected before treatment and at the first clinical control, when the patients had received rimonabant, 20 mg daily, for 4 weeks. All patients gave their informed consent and the study was performed in accordance with the Declaration of Helsinki and approved by the Regional Ethical Committee in Stockholm.

### Flow cytometry

The phenotypes of cells in the blood were analyzed by flow cytometry according to standard procedures at the Hematopathology Unit, Dept. of Pathology, Karolinska University Hospital, using 6 color flow cytometry to detect T, B, NK cells and CD3− CD4+ cells (monocytes and dendritic cells). Flow cytometry was performed on a CANTO 1 flow cytometer (BD, Becton-Dickinson, Europe).

For data acquisition and analysis, a CANTO 1 flow cytometer (BD, Becton Dickinson, Europe) was used with Cell Quest software (Becton Dickinson, Franklin Lakes, New Jersey, USA). All samples were analyzed by setting appropriate side and forward scatter gates to identify the mononuclear cell population, using CD45 and forward and side scatter for gate setting. Consistency of analysis parameters was ascertained by calibrating the flow cytometer with calibrating beads and FacsComp software, both from Becton Dickinson. The results are reported as percentage of gated cells positive for each antibody. The following fluorochrome conjugated antibodies, all from BD, were used: CD4 PE, CD3 PerCP-Cy5.5, CD19 PE-Cy7, CD8 APC and CD45 APC-H7. We also used BD Multitest 6-Color TBNK Reagent containing CD3 FITC clone SK7, CD16 PE clone B73, CD56 PE clone NCAM 16.2, CD45 PerCP-Cy5.5 clone 2D1, CD4 PE-Cy7 clone SK3, CD19 APC clone SJ25C1 and CD8 APC-Cy7, clone SK1. The gating strategy is shown in Fig. S1.

### RNA isolation and oligonucleotide array hybridization

Blood mononuclear cells were isolated by Ficoll separation (Ficoll-Paque PLUS, GE Healthcare, Little Chalfont, UK). From the cell-pellet total RNA was prepared using Qiagen midi plus kit (Qiagen GmbH, Hilden Germany) as recommended by the manufacturer and was quality controlled on an Agilent Bioanalyzer (Agilent Technologies, Inc. Palo Alto, California, USA). Six pretreatment samples and seven samples obtained after rimonabant treatment passed the quality control. The cRNA synthesis for microarray experiments and the hybridizations were carried out using Affymetrix Human Gene 1.0 ST Array (Affymetrix, Inc., Santa Clara, California, USA) according to standard Affymetrix

Almestrand et al. (2015), *PeerJ*, DOI 10.7717/peerj.1056

**Table 1 Clinical parameters of included subjects and percentages of blood cells as analyzed by flow cytometry before and after 4 weeks of rimonabant treatment.** Total T cells were defined as CD3+, CD4+ T cells as CD3+CD4+, CD8+ T cells as CD3+CD8+ and B cells as CD19+. The CD3−CD4+ cell population consists of monocytes and dendritic cells. The CD3−, CD16+ and/or CD56+ contain NK cells and monocytes with CD16 expression. There was a significant increase in CD3−, CD16+ and/or CD56+ cells after treatment ($p = 0.049$, paired $t$-test), all other changes were non significant.

| Patient | Age, sex | Weight change (kg) | Leukocyte subsets as analyzed by flow cytometry[*] | | | | | | | | | | | |
| --- | --- | --- | --- | --- | --- | --- | --- | --- | --- | --- | --- | --- | --- | --- |
| | | | Total T cells | | CD4+ T cells | | CD8+ T cells | | B cells | | CD3–CD4+ cells | | CD3−, CD16+ and/or CD56+ | |
| | | | Before | After | Before | After | Before | After | Before | After | Before | After | Before | After |
| 1 | 50, F | nd | 75 | 73 | 61 | 59 | 14 | 13 | 15 | 12 | 7.2 | 8 | 9 | 14 |
| 2 | 41, F | −6.3 | 58 | 61 | 37 | 34 | 21 | 26 | 32 | 25 | 7 | 6.6 | 8 | 12 |
| 3 | 55, F | nd | 72 | 72 | 54 | 53 | 19 | 20 | 16 | 12 | 5.7 | 8 | 11 | 15 |
| 4 | 56, M | −3.1 | 70 | 72 | 48 | 51 | 21 | 20 | 15 | 17 | 5.9 | 4.6 | 14 | 10 |
| 5 | 47, F | −2.8 | 75 | 73 | 51 | 49 | 23 | 24 | 11 | 12 | 4.9 | 8 | 13 | 15 |
| 6 | 44, F | 0.0 | 69 | 70 | 51 | 50 | 17 | 19 | 20 | 18 | 3.9 | 4.6 | 8 | 12 |
| 7 | 69, F | −2.0 | 81 | 81 | 37 | 35 | 45 | 46 | 8 | 8 | 3.3 | 3.4 | 9 | 11 |
| 8 | 58, M | −3.0 | 85 | 87 | 52 | 53 | 32 | 40 | 4 | 3 | 2.1 | 5.7 | 7 | 9 |
| Median | 52.5 | 2.9 | 73.5 | 72.5 | 51 | 50.5 | 21 | 22 | 15 | 12 | 5.3 | 6.15 | 9 | 12 |
| (range) | (41–69) | (0–6.3) | (58–85) | (61–87) | (37–61) | (34–59) | (14–45) | (13–46) | (4–32) | (3–25) | (2.1–7.2) | (3.4–8) | (8–14) | (9–15) |

**Notes.**

[*] Values are percentage of cells in mononuclear gate.

protocols at the core facility for Bioinformatics and Expression Analysis, Department of Biosciences and Nutrition, Karolinska Institutet.

## Gene expression data analysis

We used tools provided in the Partek Genomic Suite 6.5 software (Partek Inc., St. Louis, Missouri, USA). Normalization was done by Robust Multiarray Analysis (RMA) followed by 1-way Analysis Of Variance (ANOVA) comparing the patient group before and after treatment. Significantly changed genes and exons were selected with an unadjusted $p$-value of $<0.001$, a False Discovery Rate (FDR) $<0.1$ and a fold-change equal or greater than $>1.5$ for up regulated genes and equal or less than $<-1.5$ for down regulated genes. Gene functional annotations were performed by using the free software **DAVID** v6.7 (Database for **A**nnotation, **V**isualization and **I**ntegrated **D**iscovery) (*Huang da, Sherman & Lempicki, 2009*). The gene expression data are deposited at the GEO repository under the number GSE68055.

## Statistical analysis

Leukocyte subsets (as measured by flow cytometry) in blood before and after rimonabant treatment were analyzed using a paired $t$-test.

# RESULTS

## Analysis of PBMC by flow cytometry before and during treatment with rimonabant

Blood levels of mononuclear cells on eight obese patients were analyzed by flow cytometry before and during treatment with rimonabant. There were no significant changes in the relative frequencies of total CD3+ T cells, CD4+ T cells, CD8+ T cells, B cells or CD3–CD4+ cells (monocytes and dendritic cells) in the patients during the treatment period (Table 1, graphically presented in Fig. 1) There was however a trend towards an increase in percentage of CD3−, CD16+/and or CD56+ cells (before treatment median 9% range 7–14%; after treatment median 12% range 9–15% $p = 0.049$) (Table 1 and Fig. 1).

## Whole-transcript gene expression analysis demonstrates significant changes in genes belonging to innate immune system pathways

Treatment with rimonabant might influence the expression of genes in patient leukocytes. To explore possible differences in gene expression profiles, whole-transcript expression analysis of PBMC before and during treatment was done, using Affymetrix Human Gene 1.0 ST Arrays. 47 probe sets were significantly differently expressed during treatment with a fold change of at least 1, 5, 37 probe sets showed increased expression and 10 decreased expression, respectively (Table 2). Several of the genes with significantly increased expression after rimonabant treatment are known components of the innate immune system (as exemplified by KLRF1, LILRA2, CTSB, CD160, CD177, and LY96). KLRF1 (also named NKp80) encodes a lectin-type of receptor that is expressed on nearly all NK cells and stimulates their cytotoxicity and cytokine release (*Kuttruff et al., 2009*).

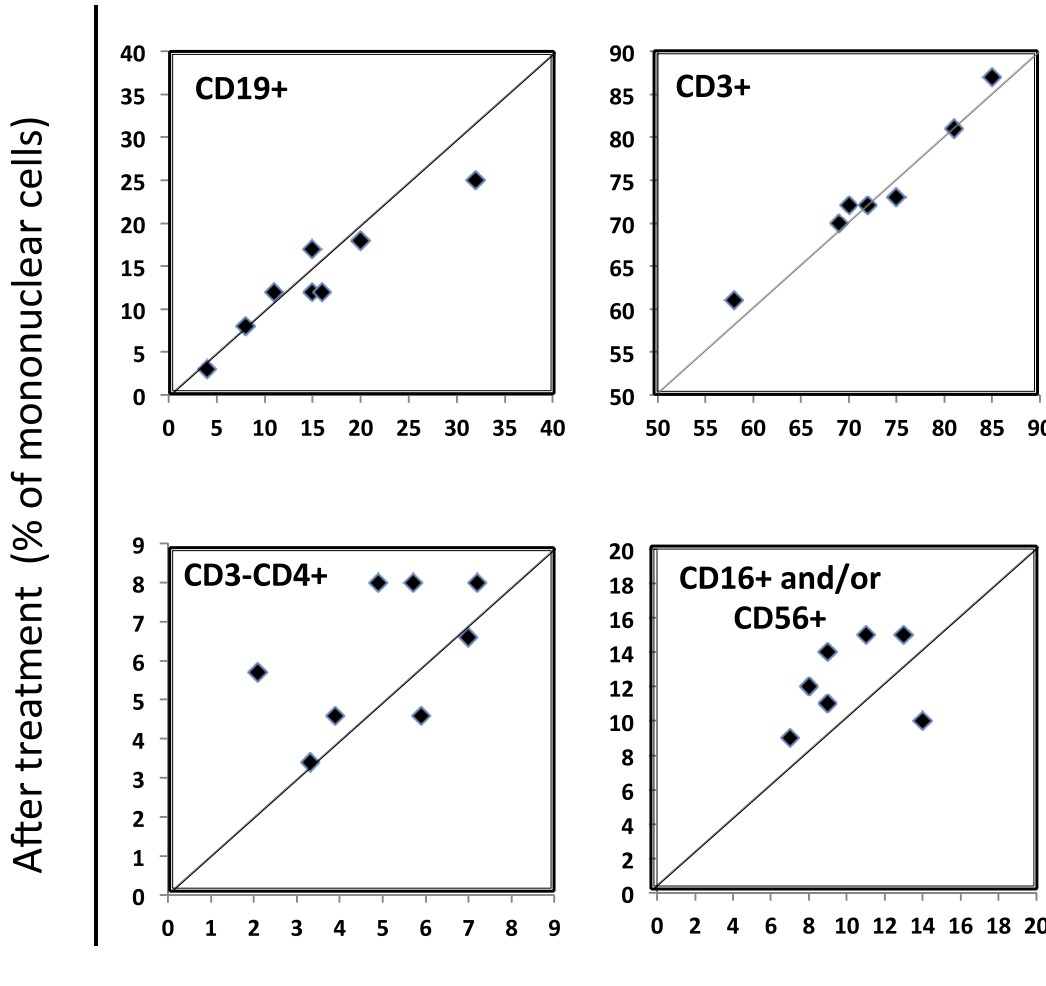

**Before treatment (% of mononuclear cells)**

**Figure 1** **Percentage of peripheral blood mononuclear cells (PBMC) before and during treatment with rimonabant.** PBMC were analyzed by flow cytometry before start of therapy and 4 weeks later and results are given as percentage of mononuclear cells in blood. Each data point represents results from one patient. In cases with no change in the frequency of a certain cell type the data point would fall on the line. The only statistically significant change was for CD3−, CD16+ and/or CD56+ cells ($p = 0.049$). For the other subsets the $p$-values were as follows: CD3+ $p = 0.47$; CD3+CD4+ $p = 0.25$; CD3+CD8+ $p = 0.11$; CD19+ $p = 0.13$; CD3–CD4+ $p = 0.11$. The mononuclear gate was defined by CD45 in combination with side and forward scatter. Within this gate the frequencies of CD3+ T cells, CD19+ B cells, CD3−, CD56+ and/or CD16+ cells (NK cells and subpopulation of CD16+ monocytes) and CD3−, CD4+ cells (monocytes and dendritic cells) were analyzed.

LILRA2 is the gene for an immune receptor that is expressed on monocytes, B cells, NK cells and dendritic cells and affects antigen presentation and innate immune responses (*Lu et al., 2012*). CTSB encodes cathepsin B, a protein that can be expressed in several immune cells including monocytes and that is involved in cell migration and immune modulation (*Staun-Ram & Miller, 2011*). CD160 is an essential NK cell receptor and is

**Table 2 Genes differentially expressed in PBMC after treatment with rimonabant (ratio >1.5 between rimonabant treated and controls, *p*-value <0.001, false discovery rate <0.1).**

| Probe set ID | Gene symbol | Fold change | *p*-value | Gene name |
|---|---|---|---|---|
| 7953892 | KLRF1 | 2.50 | 0.00046 | Killer cell lectin-like receptor subfamily F, member 1 |
| 7983910 | AQP9 | 2.25 | 0.00033 | Aquaporin 9 |
| 8031207 | LILRA2 | 2.10 | 0.00059 | Leukocyte immunoglobulin-like receptor, subfamily A |
| 8078008 | LSM3 | 2.00 | 0.00068 | LSM3 homolog |
| 7981290 | WARS | 1.99 | 0.00028 | Tryptophanyl-tRNA synthetase |
| 8149330 | CTSB | 1.90 | 0.00089 | Cathepsin B |
| 8127534 | C6orf150 | 1.87 | 0.00035 | |
| 7919243 | CD160 | 1.79 | 0.00096 | CD160 molecule |
| 8130732 | BRP44L | 1.74 | 0.00073 | Brain protein 44-like |
| 8110318 | PRELID1 | 1.72 | 0.00084 | PRELI domain containing 1 |
| 8003953 | PSMB6 | 1.69 | 0.00013 | Proteasome subunit, beta type, 6 |
| 8015545 | RAB5C | 1.68 | 0.00085 | RAB5C, member RAS oncogene family |
| 8133690 | MDH2 | 1.68 | 0.00081 | Malate dehydrogenase 2 |
| 8178676 | NEU1 | 1.67 | 0.00052 | Sialidase 1 |
| 7973110 | RNASE2 | 1.65 | 0.00028 | Ribonuclease, RNase A family, 2 |
| 026541 | FAM32A | 1.65 | 0.00032 | Family with sequence similarity 32, member A |
| 8004247 | C17orf49 | 1.64 | 0.00062 | |
| 8088820 | RYBP | 1.63 | 0.00076 | RING1 and YY1 binding protein |
| 8058373 | WDR12 | 1.62 | 0.00077 | WD repeat domain 12 |
| 8071119 | BCL2L13 | 1.61 | 0.00092 | BCL2-like 13 (apoptosis facilitator) |
| 8174103 | GK | 1.61 | 0.00086 | Glycerol kinase |
| 8016099 | EFTUD2 | 1.60 | 0.00079 | Elongation factor Tu GTP binding domain containing 2 |
| 7914563 | YARS | 1.60 | 0.00086 | Tyrosyl-tRNA synthetase |
| 979085 | PYGL | 1.59 | 0.00092 | Phosphorylase, glycogen |
| 8004237 | RNASEK | 1.59 | 0.00049 | Ribonuclease, RNase K |
| 7947681 | ARHGAP1 | 1.58 | 0.00025 | Rho GTPase activating protein 1 |
| 7959153 | COX6A1 | 1.57 | 0.00096 | Cytochrome c oxidase subunit VIa polypeptide 1 |
| 8017437 | FTSJ3 | 1.56 | 0.00055 | FtsJ homolog 3 |
| 8049180 | EIF4E2 | 1.55 | 0.00096 | Eukaryotic translation initiation factor 4E family member |
| 8146934 | LY96 | 1.54 | 0.00075 | Lymphocyte antigen 96 |
| 7900922 | ATP6V0B | 1.54 | 0.00064 | ATPase, H+ transporting |
| 8037913 | NAPA | 1.53 | 0.00037 | N-ethylmaleimide-sensitive factor attachment protein, alpha |
| 8037037 | ATP5SL | 1.52 | 0.00031 | ATP5S-like |
| 8016708 | LRRC59 | 1.51 | 0.00019 | Leucine rich repeat containing 59 |
| 8163383 | SUSD1 | 1.51 | 0.00062 | Sushi domain containing 1 |
| 7990151 | PKM2 | 1.51 | 0.00093 | Pyruvate kinase, muscle |
| 7978123 | PSME2 | 1.51 | 0.00065 | Proteasome activator subunit 2 |
| 8075564 | RFPL2 | −1.51 | 0.00020 | Ret finger protein-like 2 |
| 7900878 | ARTN | −1.51 | 0.00041 | Artemin |
| 8141228 | TMEM130 | −1.51 | 0.00075 | Transmembrane protein 130 |
| 8037298 | CD177 | −1.58 | 4.0e−005 | CD177 molecule |

Table 2 (*continued*)

| Probe set ID | Gene symbol | Fold change | *p*-value | Gene name |
|---|---|---|---|---|
| 8069142 | KRTAP10-4 | −1.59 | 0.00023 | Keratin associated protein 10-4 |
| 8070771 | KRTAP10-1 | −1.60 | 0.00054 | Keratin associated protein 10-1 |
| 8172713 | LOC347549 | −1.61 | 0.00061 | Hypothetical LOC347549 |
| 8075200 | RHBDD3 | −1.63 | 0.00067 | Rhomboid domain containing 3 |
| 8167575 | GAGE12B | −1.66 | 0.00028 | G antigen 12B |
| 8010901 | DOC2B | −1.82 | 0.00072 | Double C2-like domains, beta |

involved in regulation of cytokine production (reviewed in *Le Bouteiller et al., 2011*). CD177 is a GPI linked cell surface molecule that regulates activation and migration of neutrophil granulocytes (*Stroncek, 2007*). LY96 (also named MD-2) is associating with toll-like receptor 4 and is involved in signaling by LPS (*Mancek-Keber & Jerala, 2015*). A few genes promoting increased apoptosis were also upregulated (BCL-like 13, an apoptosis facilitator (*Jensen et al., 2014*; *Kataoka et al., 2001*), RING1- and YY1-binding protein, a regulator of MDM2 (*Chen et al., 2009*)).

It has previously been shown that chronic marijuana users have increased expression of CB1 in peripheral blood mononuclear cells (*Nong et al., 2002*). We therefore specifically analyzed the expression of genes belonging to the endocannabinoid system in our patient cohort. Rimonabant treatment did neither affect the expression of CB1 (mean and standard deviation of CB1 expression values before and after treatment were $12.5 \pm 3.24$ and $10.79 \pm 2.43$, respectively, corresponding to a fold change of $-1.1$) nor of CB2 or the enzymes involved in the degradation and/or synthesis of endocannabinoids (fatty acid amide hydrolase, FAAH, and N-acyl phosphatidylethanolamine phospholipase D, NAPE-PLD) either when analyzed by gene expression analysis or by RT-PCR (data not shown).

## DISCUSSION

In this study we investigated the possible effects on PBMC of treatment with the CB1 antagonist rimonabant in patients taking the drug for obesity. We found that the distribution of leukocyte subsets remained rather constant, as analyzed by flow cytometry before treatment and after 4 weeks of treatment with rimonabant with the exception of CD3−, CD16+ and/or CD56+ cells that increased after treatment. This subset includes NK cells (CD3−, CD56+ and/or CD16+) and also subsets of monocytes (CD3–CD16+). There were no significant changes in expression levels of cannabinoid receptors or enzymes involved in synthesis and metabolism of endocannabinoids. However gene expression analysis suggested that genes involved in metabolism, cell death and the innate immune system were up regulated during treatment.

Rimonabant is the first selective CB1 antagonist registered for clinical use and was clinically developed for treatment of obesity and the metabolic syndrome. Beside the effect on food intake, anti-proliferative actions on normal and malignant cells have been reported. Cannabinoid receptors are often more highly expressed on malignant cells than on their normal counterparts and cancer cells are usually more sensitive to
the action of cannabinoids than normal cells (reviewed in *Flygare & Sander, 2008*; *Sido, Nagarkatti & Nagarkatti, 2014*; *Wasik, Christensson & Sander, 2011*). Rimonabant has been reported to induce growth inhibition or apoptosis on several malignancies including breast, thyroid and colon cancer (*Bifulco et al., 2004*; *De Petrocellis et al., 1998*; *Santoro et al., 2009*; *Sarnataro et al., 2006*). We have previously demonstrated that mantle cell lymphoma and other B cell lymphomas have higher expression of CB1 and CB2 than normal lymphocytes (*Gustafsson et al., 2008*; *Islam et al., 2003*; *Wasik et al., 2014*). Cannabinoid receptor agonists, at 1–10 µM levels, reduced proliferation and induced programmed cell death in mantle cell lymphoma *in vitro* and in a xenotransplant model (*Flygare et al., 2005*; *Gustafsson et al., 2006*; *Gustafsson et al., 2008*; *Schatz et al., 1997*; *Wasik et al., 2011*). Interestingly, similar concentrations of rimonabant induced cell death in *ex vivo* isolated mantle cell lymphoma cells (*Flygare et al., 2005*). While these studies suggest that targeting of CB1 may be of use in cancer therapy, concern may be raised since anti-proliferative effects have been reported in PBMC (*Malfitano et al., 2008*). In these studies, rimonabant inhibited mitogen induced cell proliferation *in vitro* via G1/S phase arrest without induction of cell death (*Malfitano et al., 2008*). In contrast, *Gallotta et al. (2010)* reported that freshly isolated PBMC are highly resistant to the cytotoxic and cytostatic effects of rimonabant compared to leukemia-derived cell lines. It is possible that the different sensitivity to CB1 antagonism in freshly isolated, compared to mitogen activated, PBMC may reflect differences in expression levels of CB1. Resting leukocytes express very low levels of CB1 (*Bouaboula et al., 1993*; *Galiegue et al., 1995*; *Kaminski et al., 1992*) while receptor levels may increase upon activation by mitogens, cytokines or exposure to CB1 agonists (*Borner et al., 2007*; *Nong et al., 2002*; *Schatz et al., 1997*). We did not detect any significant differences in expression levels of CB1 or other components of the endocannabinoid system during rimonabant treatment for 4 weeks. Furthermore, our studies on *ex vivo* isolated PBMC from rimonabant treated patients demonstrated very minor changes in frequencies of T cells, B cells, CD3−, CD16+ and/or CD56+ cells or CD3–CD4+ cells or on total lymphocyte counts, in line with the results of *Gallotta et al. (2010)*.

Global gene expression analysis demonstrated significant changes in genes coding for components of the innate immune system. The study design does not make it possible to discriminate if the differences in gene expression can be ascribed to certain subsets of leukocytes or if it is a general process, seen in all PBMC. However, many of the genes that were more highly expressed after treatment with rimonabant are expressed in NK cells (such as KLRF1 and CD160) and monocytes, which imply that the treatment is associated with the activation of certain inflammatory and immunological functions of the innate immune system. Interestingly, rimonabant has been shown to directly activate human and mouse macrophages and thereby inhibit the development of the intracellular pathogen Brucella suis (*Gross et al., 2000*). Furthermore, studies on lipopolysaccharide activated human macrophages showed that CB1 receptor blockade by rimonabant suppressed production of inflammatory cytokines (IL-1$\beta$, IL-6, IL-8, TNF-$\alpha$) and matrix metalloproteinase-9 (*Sugamura et al., 2009*).

## CONCLUSIONS

In conclusion our results show that rimonabant treatment induces expression of genes involved in immune responses but have only marginal effects on leukocyte subset frequencies in blood. This is in marked contrast to previous studies in which rimonabant induced cell death in malignant B lymphocytes that express high levels of CB1 (*Flygare et al., 2005*). The relatively small effects on normal leukocytes suggest that CB1 targeting may be further investigated as a therapeutic approach in lymphoma treatment, enabling selective effects of tumor cells.

### Funding

This study was supported by grants from the Swedish Cancer Society, the Swedish Research Council, the Cancer Society in Stockholm, Karolinska Institutet Funds and Stockholm County Council funds. The funders had no role in study design, data collection and analysis, decision to publish, or preparation of the manuscript.

### Grant Disclosures

The following grant information was disclosed by the authors:
The Swedish Cancer Society.
The Swedish Research Council.
The Cancer Society in Stockholm.
Karolinska Institutet Funds.
Stockholm County Council.

### Competing Interests

Stefan Almestrand is an employee of AstraZeneca. The authors declare there are no competing interests.

### Author Contributions

- Stefan Almestrand performed the experiments, analyzed the data, prepared figures and/or tables, reviewed drafts of the paper.
- Xiao Wang, Åsa Jeppsson-Ahlberg, Marcus Nordgren and Jenny Flygare performed the experiments, analyzed the data, reviewed drafts of the paper.
- Birger Christensson conceived and designed the experiments, performed the experiments, analyzed the data, contributed reagents/materials/analysis tools, wrote the paper, reviewed drafts of the paper.
- Stephan Rössner conceived and designed the experiments, performed the experiments, analyzed the data, contributed reagents/materials/analysis tools, reviewed drafts of the paper.
- Birgitta Sander conceived and designed the experiments, performed the experiments, analyzed the data, contributed reagents/materials/analysis tools, wrote the paper, prepared figures and/or tables, reviewed drafts of the paper.

## Human Ethics

The following information was supplied relating to ethical approvals (i.e., approving body and any reference numbers):

All patients gave their informed consent and the study was performed in accordance with the Declaration of Helsinki and approved by the Regional Ethical Committee in Stockholm. The ethical number is Dnr_1267_31.

## Microarray Data Deposition

The following information was supplied regarding the deposition of microarray data:

The gene expression data are deposited at the GEO repository under the number GSE68055.

## Supplemental Information

Supplemental information for this article can be found online at http://dx.doi.org/10.7717/peerj.1056#supplemental-information.

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
