# Peer review of "Influence of rimonabant treatment on peripheral blood mononuclear cells; flow cytometry analysis and gene expression profiling"

_PeerJ, doi:10.7717/peerj.1056_

## Round 0.1 · original submission · Major Revisions

Dear Dr. Sander,

You have addressed relevant and timely topics, however the reviewers have raised number of technical concerns. If you decide to address all the critical point experimentally and/or theoretically, please, provide point-by point explanation or each criticism of the reviewers in the accompanying letter.

Reviewer 1 ·

Basic reporting

Accession number to deposited microarray data was not provided.

References and in-text-citations should be adjusted to journal style according to Instructions for Authors

Experimental design

No comments

Validity of the findings

No comments

Additional comments

1. The major concern about the study is a phenotyping of blood monocytes using only CD3 and CD4 markers (although some previous studies used these markers to describe monocytic cell population). In my opinion, CD14 marker is necessary to identify monocytes since CD3-CD4+ phenotype may be observed in other cells (e.g. pDC).
2. Please provide example of flow cytometry gating for mononuclear cell phenotyping
3. In the Results section authors describe that no significant differences were found for total CD3+ T cells, CD4+ T cells, CD8+ T cells, B cells and monocytes. Please provide p-values for each cell type.
4. Please provide the data about expression levels of CB1 receptor on different cell types or whole PBMCs before and after treatment with rimonabant.
5. In the Methods section it is written that both CD16 and CD56 abs were labeled with PE. Since NK cells were identified as CD16+CD56+ double-positive cells it is unclear how authors differentiated between these markers.
6. Authors use term "Antagonist" for rimonabant in the Abstract and "Inverse agonist" in other sections. The difference exists between these terms. Please define what is more proper in case of rimonabant.
7. The text of the manuscript has to be checked for typing mistakes

Reviewer 2 ·

Basic reporting

In this work “Influence of rimonabant treatment on peripheral blood mononuclear cells; flow cytometry analysis and gene expression profiling” by Almestrand et al., the authors investigated possible effects on peripheral blood mononuclear cells (PBMC) in patients treated with rimonabant. Then, they evaluated leukocyte subsets by 6 color flow cytometry in eight patients before and at treatment with rimonabant for 4 week and whole-transcript gene expression profiling in PBMC before and at 4 weeks of rimonabant treatment. They evidence no significant changes of monocytes, B cells, total T cells or T cell subsets in PBMC during treatment with rimonabant, but only a small but significant increase in NK cells after rimonabant therapy. Finally, they observed by gene expression analysis, changes in expression of genes associated with innate immunity, cell death and metabolism.
It is not clear how the aim of the study is connected with the background. The author highlight in the background how rimonabant can exert anti-proliferative effects in malignant lymphoma cells, on activated normal lymphocytes in vitro, two systems that are not “ normal “, while in another study performed on a “normal “system, non-activated PBMC, the anti-proliferative effects are not evident. In their study, they recruited obese patients treated with rimonabant and isolate the blood of these patients to see differences in leukocyte subsets, the authors should add a link between obesity and immune cells to better correlate the aim of the study with the background.

Experimental design

It is not surprising that there are not differences in the frequency of monocytes, B cells, total T cells, or T cell subsets in PBMC, because the CB1 is not primarily expressed in these cells. The author also state in the last sentences “to investigate which gene were differentially expressed during CB1 receptor blockade” but the author do not prove that rimonabant actually acts via CB1 in this system. To this aim, they should use an antagonist to show that the regulation of the expression of those genes was regulated by the CB1 or in alternative, a method to silence the CB1 receptor in these cells?

Concerning the increment they observe in the NK cells after the treatment with rimonabant is it significant? Can they exclude that the patients have contracted any infections that could alter the immune cell asset and so compromise their results?

Validity of the findings

The authors highlight some genes that are regulated by rimonabant treatment, they should provide more information about these genes, however the gene LILRA1 is not reported in the table, please explain.
The authors list CD177 among the up regulated genes but the fold change is -1,58 , it is not very much up-regulated, please explain.

Additional comments

In the conclusion, the authors mark differences between their study on blood isolated from obese patients and treated with rimonabant, with cell death induced by rimonabant in malignat lymphoma cells, these are different systems, what is the connection?
Finally, they conclude their study, (see conclusion), suggesting further investigation of CB1 targeting in lymphoma treatment, but I do not see how their study can support this conclusion.
Minor revision
There are several grammar errors throughout the manuscript, as example:
Material and methods, Flow cytometry, line 80 “ The phenotypes of cells…was” it should be “ the phenotype”.
Results, line 141 “ It has previously has been shown” please revise this sentence.

---

## Round 0.2 · Minor Revisions

Dear Authors,

Please, address the comments of the reviewer about the gating strategy for CD3-CD4+ population in the supplementary figure and comment on the subpopulation of CD3-CD16+ cells. It is well-known that specific subpopulations monocytes can be also CD16+ (and obviously CD3-)

Moreover CD16+ monocytes have disease-related functions and molecular profile.

Reviewer 1 ·

Basic reporting

No comments

Experimental design

No comments

Validity of the findings

No comments

Additional comments

1. Please add gating strategy for CD3-CD4+ population as a supplementary figure
2. Authors should be careful with definition of cell phenotypes. As I understood from the revised version of the manuscript NK cells were defined as CD3-CD56+ or CD3-CD16+ cells. However, subset of "inflammatory" monocytes in the blood also expresses CD16. Please clarify

Reviewer 2 ·

Basic reporting

No comment

Experimental design

No comment

Validity of the findings

No comment

Additional comments

No comment

---

## Round 0.3 · accepted · Accept

Dear Authors, thank you for addressing the critical points of the reviewers properly. Your subimssion is accepted for the publication.
Thank you for the contribution to PeerJ

With best regards

Julia Kzhyshkowska

Reviewer 1 ·

Basic reporting

No comments

Experimental design

no comments

Validity of the findings

no comments